# The Synergistic Effect of Polystyrene/Modified Boron Nitride Composites for Enhanced Mechanical, Thermal and Conductive Properties

**DOI:** 10.3390/polym15010235

**Published:** 2023-01-02

**Authors:** Shafi Ur Rehman, Sana Javaid, Muhammad Shahid, Nasir Mahmood Ahmad, Badar Rashid, Caroline R. Szczepanski, Asim Shahzad

**Affiliations:** 1School of Chemical and Materials Engineering (SCME), National University of Sciences and Technology (NUST), Islamabad 44000, Pakistan; 2School of Natural Sciences (SNS), National University of Science and Technology (NUST), Islamabad 44000, Pakistan; 3Department of Chemistry, University of Wah, Quid Avenue, Wah Cantt, Rawalpindi 47040, Pakistan; 4Dean of Research and Development (R & D), National University of Technology NUTECH, Islamabad 44000, Pakistan; 5Department of Chemical Engineering & Materials Science, Michigan State University (MSU), East Lansing, MI 48824, USA; 6School of Engineering, Swansea University, Swansea SA2 8PP, UK

**Keywords:** thermal conductivity, polymeric composites, boron nitride

## Abstract

Thermal conductivity (TC) and thermal stability are the basic requirements and highly desirable properties in thermal management, heat storage and heat transfer applications. This work is regarding the fabrication of polystyrene/boron nitride composites and melt extruded to produce good thermal stability, increased thermal conductivity and enhanced mechanical properties. Our strategy is potentially applicable to produce thermally conductive composites of low cost over large scale. Boron nitride powder is bath sonicated in 10% NH_3_ solution to avoid its agglomeration and tendency toward entanglement in a polymer matrix. An approximately 67.43% increase in thermal conductivity and 69.37% increase in tensile strength as well as 56 multiple increases in thermal stability of the optimum samples were achieved. The developed polymeric composites are potentially applicable in the electronic industry, especially in electronic devices used for 5G, heat sink and several other aviation applications.

## 1. Introduction

Polymer composites are an ideal choice for materials applications due to their lighter weight, low cost, chemical inertness, thermal stability, electrical insulating properties, and good machinability. They are employed in many aspects of daily life, from electronic circuits to the aerospace industry, transport, and construction sectors, as well as for MRI scanners and surgical target tools [1,2,3,4]. Due to these benefits, mankind is currently experiencing and exploring the fourth zone of life, the “polymer zone”, after experiencing the stone, bronze, and iron ages. However, there are still several operational challenges regarding polymers that researchers are facing and addressing [5,6]. The most common disadvantages of polymer matrix composites (PMCs) are sensitivity to abrasion, low fatigue resistance, low impact strength, service temperature limit, degradation, and emission of radiation upon exposure to sunlight [7,8,9]. Apart from that, polymers also offer the disadvantage of poor heat dissipation and high thermal expansion, which act as barrier to employing PMCs in many useful manufacturing industries [10,11]. Polymers are basically comprised of organic macromolecules linked with covalent bonding, as shown in Figure 1 [12,13]. The available electrons are interlocked among these bonds and their free movement is restricted, resulting in poor conductivity. To improve this associated handicap with polymers, researchers are currently exploring new strategies for addressing these limitations. 

The inadequate thermal stability and intrinsic poor thermal conductivity of polymers are two major concerns in the field of PMCs [15,16]. To address these issues, many researchers have contributed their part by incorporating chemically stable and thermally conductive fillers, such as alumina (Al_2_O_3_), carbon nano tubes (CNTs), graphene, chalk, metal, silica, aluminum nitride (AlN) and silicon carbide in variety of polymers [17]. Additionally, a large number of other metal-based, ceramic-based, and carbon-based fillers are also used to overcome the thermal management and heat dissipation problems of PMCs [18,19,20,21,22,23]. Regular and uniform distribution of filler in a polymer matrix is an important requirement for achieving thermally conductive composites. Otherwise, agglomeration and inadequate distribution of filler in polymers result in poor conductivity. To counter this, the conductive pathways within the matrix must be established using an effective amount of filler [24,25,26]. Figure 2 illustrates the formation of both thermal pathways over adequate and inadequate amount of filler added to matrix.

However, integrating thermally conductive fillers and generating PMCs with required conductivity, is not an easy job. There are various factors effecting conductivity, such as the band gap of the employed filler, interfacial thermal resistance between the filler-matrix and filler-filler, loading level of filler, percolation threshold theory and method of composites fabrication [27,28]. The concept of percolation threshold theory was initially presented by Broadbent & Hammersley in 1957 [29]. 

Since then, this theory has been intensively studied by the researchers. The formation of these networks called “percolation” are shown in Figure 3. The fillers concentration and network formation via percolation within the matrix often brings a consequential change in mechanical and electrophysical properties of the fabricated PMCs. The concentration of filler in matrix should be adjusted accordingly to avoid the formation of under-growing or over-growing clusters. 

Several methods are being used to enhance the thermal conductivity of polymers. These methods are sometime more complex and demanding and are hard to adopt on a large scale, for example, alignment of chain orientation, controlled mechanical stretching, applying nanoscale templating, inter-chain or side-chain changes and multiple filler networks, etc. [30,31,32,33].

Polymers, such as nylon 6, 6, were made thermally and electrically conductive using various carbon-based conductive fillers by Julia et al., while Erick et al. used carbon black and graphite partials for the said purpose [34,35]. Metal oxides and glass fibers were added to polypropylene up to 50 vol% in order to achieve conductivity of 2.5 W/(m·K). BN and AlN particles are added to increase the thermal conductivity of polypropylene [36,37]. AlN powder is also added in polystyrene matrix, and product is obtained through in situ polymerization using infiltration procedure [38]. Jun-Wei Gu et al. fabricated polystyrene composites by adding several thermally conductive fillers, such as (Al_2_O_3_), (MgO) and silicon carbide particles, up to 40 vol% loadings and achieved thermal conductivity until 1.18 W/mK [39]. 

Here, we fabricated polystyrene/boron nitride composites through the melt extrusion method. To the best of our knowledge, general purpose polystyrene and BN powder composites have not been tried so far for enhancing thermal conductivity. Keeping in mind the inherent agglomeration disadvantage of boron nitride powder, washing and sonication is done with 10% NH_3_ solution. The results obtained are encouraging in terms of polymer conductivity and helpful for young researchers to apply on other comparable polymers.

## 2. Experimental

### 2.1. Materials and Methodology

Boron nitride powder (BN) ~1 μm, molecular weight 24.82, and 98% purity was provided by Sigma Aldrich, St. Louis, MO, USA. Analytical grade ammonium hydroxide (28.0–30.0% NH_3_ basis) purchased from Sigma Aldrich Chemie, Steinheim, Germany. General purpose polystyrene (GPPS-550P, MFI 3 gm/10 min) was provided by Pak Petro chemical Industries Pvt. Ltd., Karachi, Pakistan.

### 2.2. Composites Fabrication

Modified boron nitride (m-BN) along with stearic acid and antioxidant (Irganox 1010) were mixed with polystyrene in a twin-screw extruder (Modal, Thermo Haake Poly-lab Rheomix-600, Internal Mixer, Karlsruhe, Germany). After initial digestion of the ingredients at low spin (100 °C and 60 rpm), the twin-screw rollers were adjusted to optimum parameters (200 °C and 100 rpm). Di-methacrylate (DMC) was added drop-wise to facilitate the crosslinking. Formulation details for fabricated samples are provided in Table 1 and Table 2.

### 2.3. Characterization Techniques

The structural analysis of the pristine PS, BN, m-BN and their polymeric composites were performed using Fourier Transform Infrared Spectroscopy (Nicolet 6700, Thermo Scientific, Waltham, MA, USA). Samples were analyzed in a range of 4000 to 500 cm^−1^ wavenumbers versus percent transmittance, with an average of 16 scans performed for each sample. Surface morphology of the pristine sample and the impact of adding filler were examined through SEM (Modal JSM 6490LA, JEOL, Tokyo, Japan) at 20,000 V. Specimens were sputtered and coated with gold to observe the cross-sectional view of dispersed particles inside the polymer matrix. Mechanical properties were determined using a tensile testing machine (modal TGI 5kN, Novi, MI, USA). ASTM D-638 was followed and specimens were cut into dumbbell shapes in 1mm thick compression molded sheets (Dimension: Type 4, Standard: ISO 37:1994). The data of modified samples were recorded in terms of tensile strength (TS), Young’s modulus (E) and elongation at break (Eb). Thermal resistance of the pristine PS and modified samples after adding filler content was studied through thermogravimetric analysis TGA (Model SDTA851e, Schwerzenbaclz, Switzerland). An approximately 5 mg sample was placed in an alumina crucible and the temperature was raised to 600 °C at a rational increase of 10 °C/min. An inert environment containing pure nitrogen gas (N_2_ = 99.999% purity, Grade 5) was used. TGA data is presented by plotting a graph as a function of temperature versus weight loss percentage. Differential scanning calorimetry (Thermo DSC Q 100 Thermo Model 970701.901 DSC System, MI, USA) was performed to study the thermal properties (Tg, Tc and Tm) of pristine PS and composite samples. The samples for DSC were cut into small pieces and ~5 mg of each sample was used for analysis. The sample was heated from 25 °C to 250 °C at a heating rate of 10 °C/min under a nitrogen atmosphere. Dynamic mechanical analysis (Discovery Model DMA 850, New Castle, DE, USA) was used to extract curves of storage modulus (E′), loss modulus (E″) and damping factor (tan δ) for pristine PS and composite samples, after loading filler content. Three-point bending moment was observed at a fixed frequency of 1 Hz and temperature was scanned from 25 to 150 °C at a rate of 5 °C/min. Thermal conductivity of the samples was investigated through a thermal conductivity analyzer (NETZSCH, Model LFA 447 NanoFlash^®^, MI, Selb, Grmany). The square samples of 12.7 mm (Length × Width) were graphite coated and thermal conductivity was calculated at 25 °C, 75 °C, 100 °C and 150 °C.

## 3. Results and Discussion

### 3.1. Fourier Transform Infrared Spectroscopy (FTIR)

FTIR analysis of pristine boron nitride (BN) and modified boron nitride powder (m-BN) was studied in the range of 4000–500 cm^−1^ to identify the functional group interactions after modification of boron nitride, as shown in Figure 4. The spectra of pristine (BN) clearly shows the B-N stretching at B-N-B out-of-plane bending at 1400 cm^−1^ and a sharp peak between 700–800 cm^−1^ [40]. While a broad band at 3400 cm^−1^ confirms the successful modification of the O-H group on BN, thus in the spectra of m-BN there is a slight increase in broadness in the range of 3400 to 3000 cm^−1^ [41]. FTIR spectra of pristine polystyrene (SN0) and PS/m-BN composites (SN10, SN20 and SN30) are shown in Figure 5. Sample (SN0) spectra shows the characteristics peaks of C-H stretching in the range of 2900 to 2800 cm^−1^ and C=C stretching at 1600 cm^−1^, attributed to the benzene ring, respectively [41]. SN10, SN20 and SN30 spectra show the same characteristic peaks of C-H and C=C stretching vibrations in addition to B-N bending vibration at 810 cm^−1^ in the fingerprint region, which confirms the interaction of modified boron nitride with polystyrene at the interphase of matrix and reinforcement. It is quite vivid from the spectra that peaks appear in the fingerprint region attributed to the B-N functional group and become prominent with increasing filler content up to 30 wt.%. 

### 3.2. Scanning Electron Microscopy (SEM)

Scanning electron microscopy was employed to study the surface morphology of pristine PS and PS/m-BN composites and to examine the dispersion and compatibility of m-BN as a filler in the PS matrix (Figure 6). Compared to pristine PS (SN0—Figure 6a), small, dispersed particles appear in SN10, SN20 and SN30 (Figure 6b–d), revealing good interfacial interaction of the modified BN at the matrix filler interphase. SN10 with 10 wt.% and SN20 with 20 wt.% m-BN reveals uniformly dispersed filler nanoparticles within the composites. However, with the addition of 30 wt.% m-BN, agglomeration and voids appeared to some extent due to incomplete saturation at the interphase of organic and inorganic molecules. This might be due to a difference in surface energy and polarities between the organic and inorganic phases [41]. However, with the increase of filler content (m-BN), the miscibility of the particles also increases, resulting in good adherence with the matrix that provides good mechanical and thermal properties [42]. 

### 3.3. Mechanical Analysis

Mechanical properties, including tensile strength (TS), Young’s modulus (E) and elongation at break (Eb) for pristine PS and PS/modified-BN composites, are summarized in Table 3, and Figure 7a highlights the comparison amongst tensile properties of SN0, SN10, SN20 and SN30. Pristine PS has a TS of 39.74 MPa, which is significantly increased to 57.28 MPa with the addition of 30 wt.% modified BN in SN30. These results are associated with the dispersion of m-BN nanoparticles to induce improved compatibility at the matrix-filler interphase until a susceptible limit. The comparison between Young’s modulus (E) of pristine and fabricated composite samples are shown in Figure 7b. The graph shows a maximum value of 3.65 GPa with 30 wt.% loading of modified BN as a filler in the polymer matrix. Therefore, addition of m-BN as a filler in the polymer matrix resulted in improved mechanical performance. In comparison to tensile strength and Young’s modulus, elongation at break values reduced from 3.56% to 1.88% with the incorporation of 30 wt.% modified BN as compared to the pristine. This decrease in Eb indicates reduced ductility and improves the load bearing ability without fracture and deformation. Thus, the fabricated polymeric composites presented here exhibited good mechanical properties and toughness suitable for thermal conductivity. 

### 3.4. Thermal Gravimetric Analysis (TGA)

Thermal stability of pristine and modified samples was studied by thermogravimetric analysis, as shown in Figure 8. One-fold degradation for both pristine and modified samples was completed at approximately 450 °C. It was observed from the graph that 0 wt.% remained of SN0 but increases in remaining mass were significant (8, 15 and 28 wt.%) with the addition of 10, 20 and 30 wt.% modified-BN as filler. These results demonstrate the improved thermal stability afforded by incorporating m-BN at the matrix-filler interphase. Thus, improved thermal stability enhances the mechanical properties of modified polymeric composites as evidenced from tensile data. 

### 3.5. Differential Scanning Calorimetry (DSC)

DSC was used to observe the phase transition behavior and difference in Tg of pristine PS and PS/m-BN composites due to loading of filler, as shown in Figure 9. It is observed from the graph that Tg of SN10, SN20 and SN30 composites shows a slight increase as compared to pristine SN0 due to the loading of m-BN. These results indicate better thermal stability of polymer composites due to m-BN filler and that the filler modifies the phase transition behavior (Tg) of PS composites by hindering molecular movement with flow of heat [43]. These results are attributed to strong intermolecular attractions at polymer-matrix interphase in the semicrystalline structure. Therefore, loading of m-BN into PS reduces the rate of displacement and increases the Tg, which synergistically improves the thermal property and crystallinity of the modified composites.

### 3.6. Dynamic Mechanical Analysis (DMA)

Dynamic mechanical analysis (DMA) was used to measure storage modulus (E′), loss modulus (E′′) and damping factor (tan δ) in accordance with change in time, temperature, and frequency [44]. Storage modulus (E′) in polymer composites states the resistance in deformation with respect to applying force at a low frequency. It measures the energy and force in response to elastic stress and strain on material [45]. Thus, E′ typically describes the stiffness and resistance in mobility of polymer composites with an applied elastic force. Figure 10a represents the ternary phase transition of pristine and modified composites in view of storage modulus (MPa) versus temperature at a fixed frequency of 1 Hz. At a cold, frigid temperature, the value of E′ is high due to rigidity and stiffness, while it reduces as it approaches its Tg and then remains almost constant. High stiffness and rigidity determine the resistance in movement among molecules. It was observed that hindrance in elastic movement significantly increases with m-BN filler loading and then decreases at transition stage up to Tg, which allows significant movement among the chain molecules with elevated temperature, while beyond Tg shows minor change in E′. These results confirm the marked difference in viscoelastic behavior of polymeric composite due to reinforcement at interphase. It was further observed that optimum loading of 30 wt.% filler slightly increased the storage modulus (E′) beyond the Tg of polymer composite as compared to pristine PS, which depicts the crystalline behavior and stability as evidenced by the tensile strength analysis. Beyond Tg, E′ decreases with an increase in temperature due to increased mobility and less intermolecular interaction at the matrix-filler interphase. 

Figure 10b shows the response of pristine PS and PS/m-BN composites in terms of loss modulus (E′′) versus temperature. Energy loss in terms of E′′ is initially less at low temperature for both pristine and m-BN composites due to better interfacial adhesion. It was observed that E′′ increases up to the transition phase at Tg with the maximum loading of 30 wt.% modified boron nitride. This increases in E′′ value with higher loading content revealed better crystallinity and improved intermolecular adhesion because of reinforcement and creates hindrance in the flow of material. A greater hindrance in material flow causes dissipation of energy by applying force. After transition in rubbery phase the E′′ showed a marked decrease due to viscous behavior. 

The tan δ factor depicts a ratio between E′ and E′′, representing the energy storage and loss in a cyclical manner upon loading. Tan δ as dependent variable was plotted versus temperature for pristine PS and PS/m-BN composites, as shown in Figure 10c. A three-fold transformation for pristine and modified samples was observed from the graph within the wide temperature range, following the viscoelastic behavior of polymeric composites. A strong sharp peak for pristine PS was observed in Figure 10c, while it became shortened in height with the loading of filler content. Thus, modified samples show reduced damping that was attributed to better compatibility at the interphase of polymer and filler in the polymeric composites. There was also observed an initial low damping in the glassy phase, which increased up to Tg in the glass transition phase, while reaching zero in the rubbery phase. Maximum damping was observed at Tg, illustrating the transformation of mechanical energy into thermal energy, while in the glassy state the resistance in the motion among molecular chains was restricted [46]. It was further observed that with a loading content of 30 wt.% in SN30, the damping reduces significantly as compared to pristine PS due to better crystalline structure and intermolecular adhesion at the matrix-filler interphase. 

### 3.7. Thermal Conductivity Analysis

The thermal conductivity (TC) of the fabricated samples is listed in Table 4 and shown in Figure 11. It is clear from the obtained data that thermal conductivity increased after addition of m-BN powder. This increment in TC as a result of filler addition is more prominent (1.48× greater) at room temperature (25 °C). It is observed that the TC of the samples increases gradually until 20% m-BN powder addition. This is due to the presence and mediation of the polymer among the adjacent channels of BN filler, which results in interfacial mismatch of filler and matrix, phonon scattering and ultimately lower TC [26,47]. However, increasing the amount of filler from 20% and above, a significant increase in TC is exhibited. This is probably a true reflection of the percolation threshold phenomena and formation of the m-BN networks within the matrix. Contrary to the above trend, TC of the samples with elevated temperatures (100 & 150 °C) decreases steadily until 20% filler addition. However, further addition of m-BN to 30% increases the TC of the sample.

## 4. Conclusions

General purpose polystyrene (GPPS-550P) is utilized for fabricating thermally conductive composite. Boron nitride powder is modified by washing with 10% NH_3_ solution and melt blended as filler with PS. As a result, an increase of 67.43% in thermal conductivity and 69.37% in tensile strength as well as 56 multiple increases in thermal stability were successfully achieved. The thermal conductivity of the fabricated composites at room temperature increased from 0.147 W·m^−1^·K^−1^ to 0.218 W·m^−1^·K^−1^. The fabricated composite is cheap and feasible and can be utilized as a functional polymer in any industry where thermal management is required. 

## Figures and Tables

**Figure 1 polymers-15-00235-f001:**
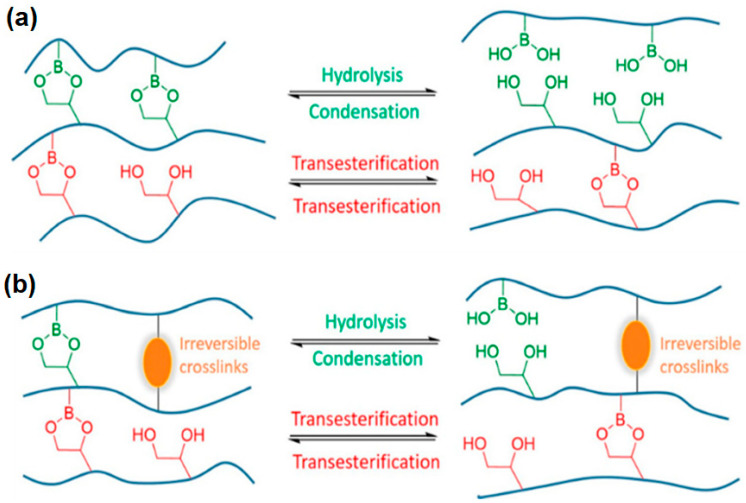
Representation of interlocked molecules in covalent bonding [14]. (**a**) Boronic esters undergo two distinct exchange mechanisms under these conditions. (**b**) When the DDN is reinforced with additional static crosslinks, its dimensional stability is enhanced and properties like creep resistance are improved.

**Figure 2 polymers-15-00235-f002:**
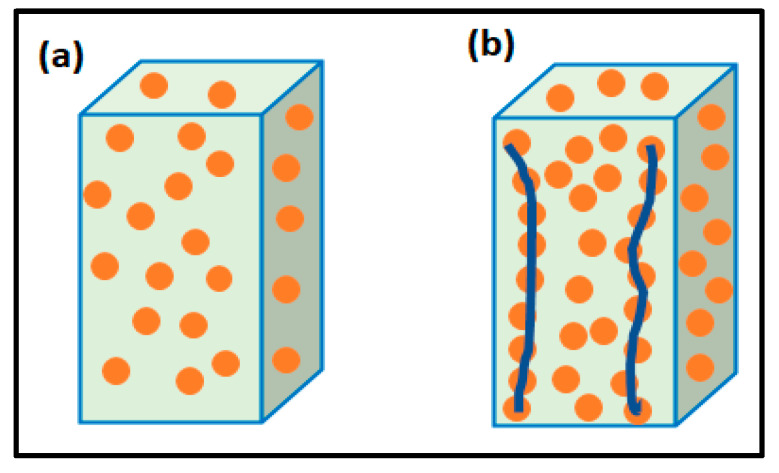
(**a**) Schematic diagram at inadequate fillers loading; (**b**) Formation of the thermal pathways at adequate fillers loading.

**Figure 3 polymers-15-00235-f003:**
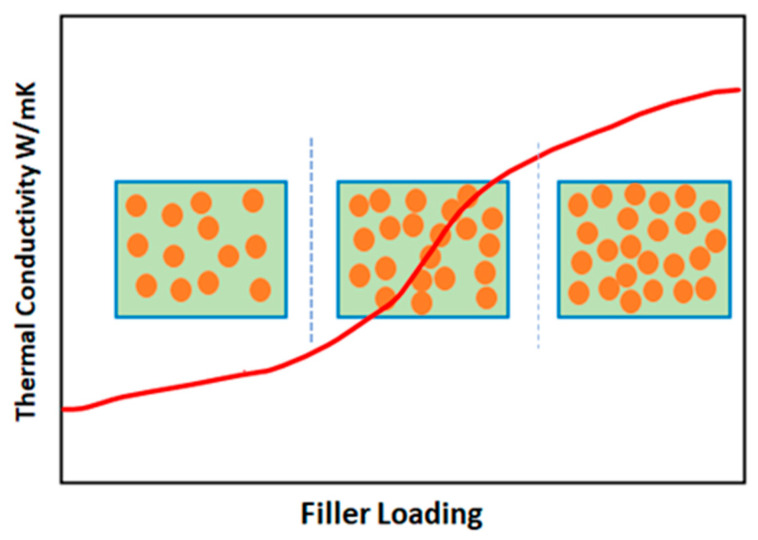
Schematic illustration of the percolation phenomenon.

**Figure 4 polymers-15-00235-f004:**
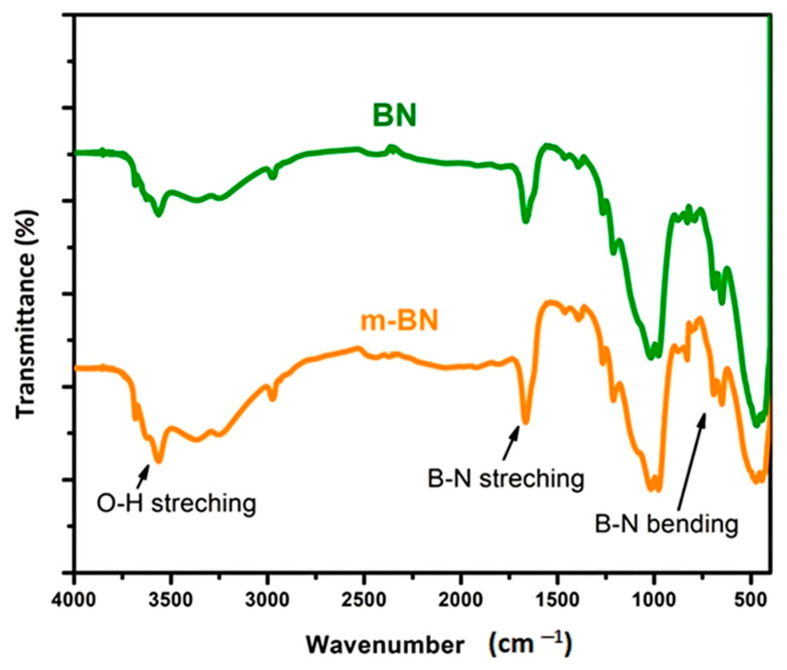
FTIR of pristine and m-BN powder.

**Figure 5 polymers-15-00235-f005:**
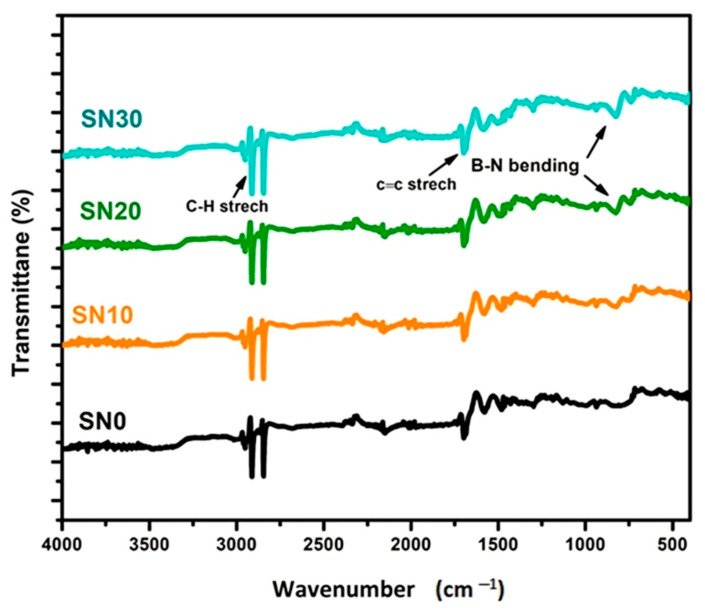
FTIR of pristine PS and PS/m-BN composites, as SN10, SN20 and SN30.

**Figure 6 polymers-15-00235-f006:**
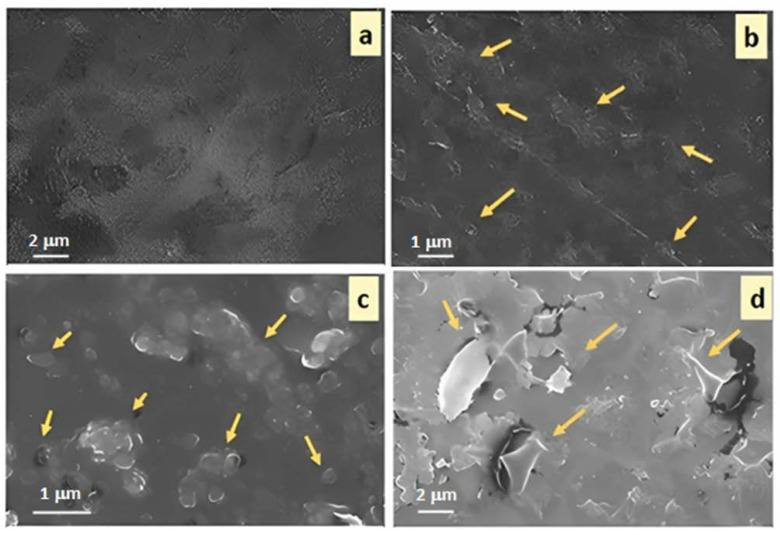
SEM micrographs in (**a**) Pristine PS (SN0), and PS/m-BN composites, as (**b**) SN10 in, (**c**) SN20 in and (**d**) SN30 in.

**Figure 7 polymers-15-00235-f007:**
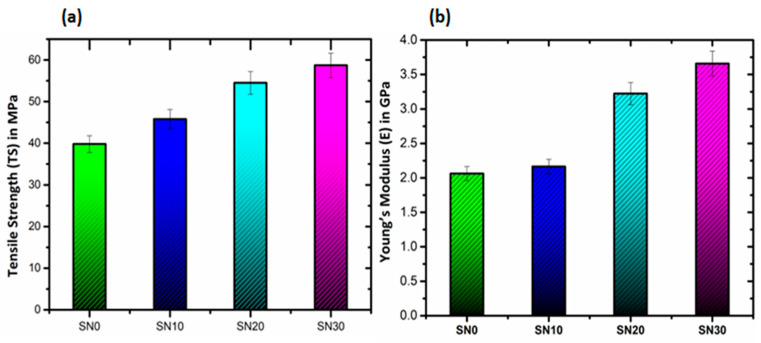
Tensile properties of PS and PS/m-BN composites are shown in (**a**) Tensile strength (TS) in MPa and in (**b**) Young’s modulus (E) in GPa.

**Figure 8 polymers-15-00235-f008:**
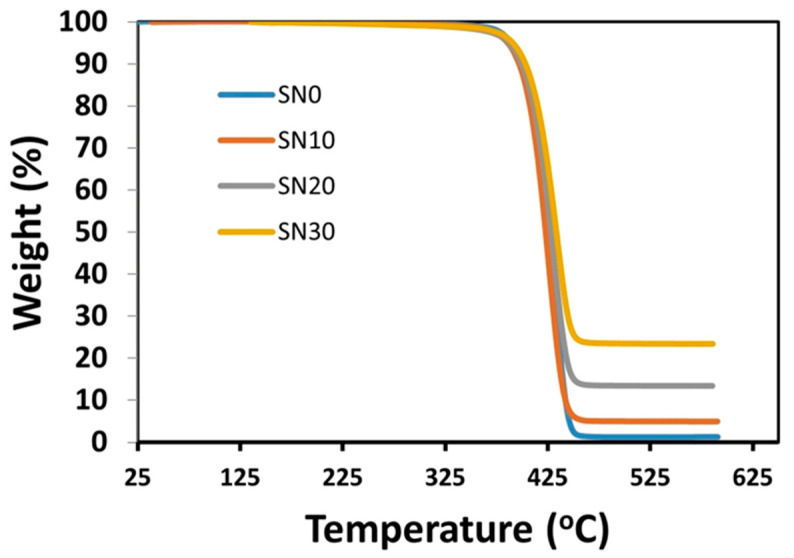
Thermogravimetric curves of pristine PS and PS/m-BN composites.

**Figure 9 polymers-15-00235-f009:**
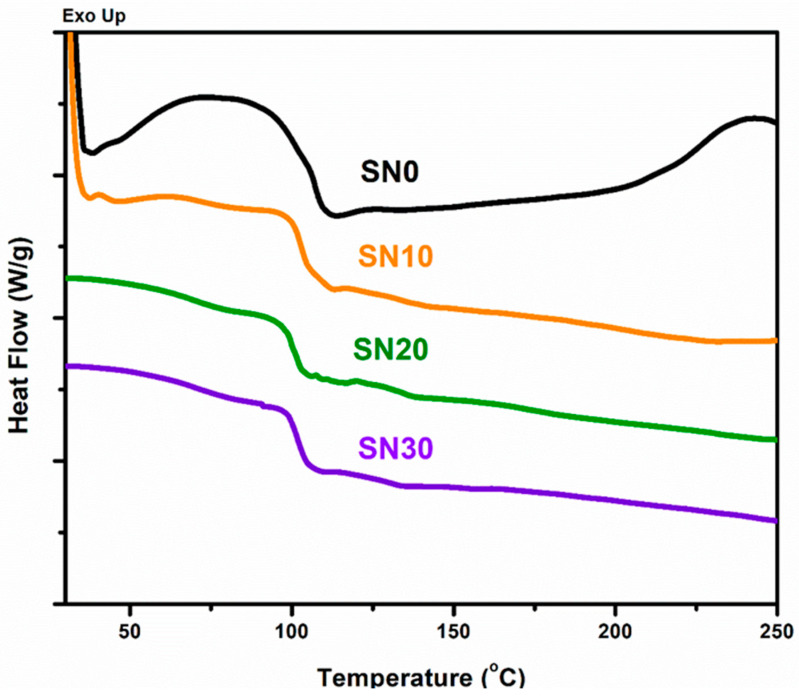
DSC curves of Pristine PS and PS/m-BN composites.

**Figure 10 polymers-15-00235-f010:**
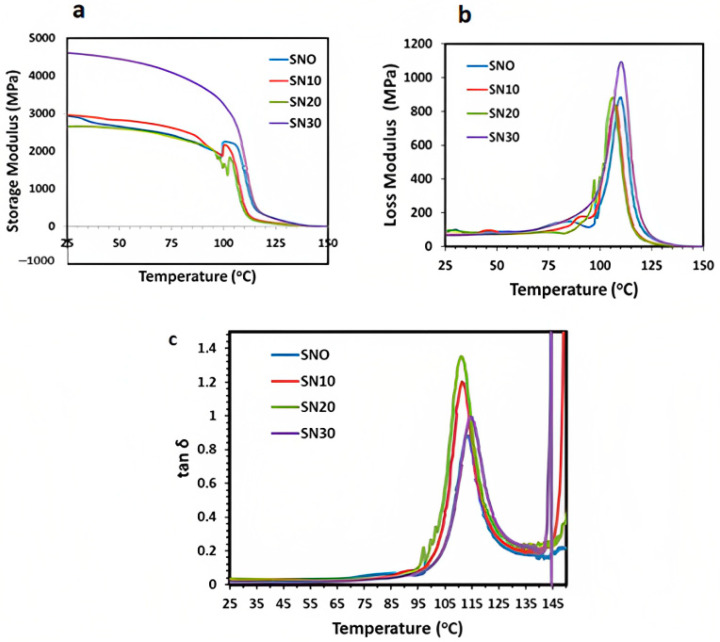
DMA curves of pristine PS and PS/m-BN composites (**a**) storage modulus (E′), (**b**) loss modulus (E′′), and (**c**) damping factor (tan δ).

**Figure 11 polymers-15-00235-f011:**
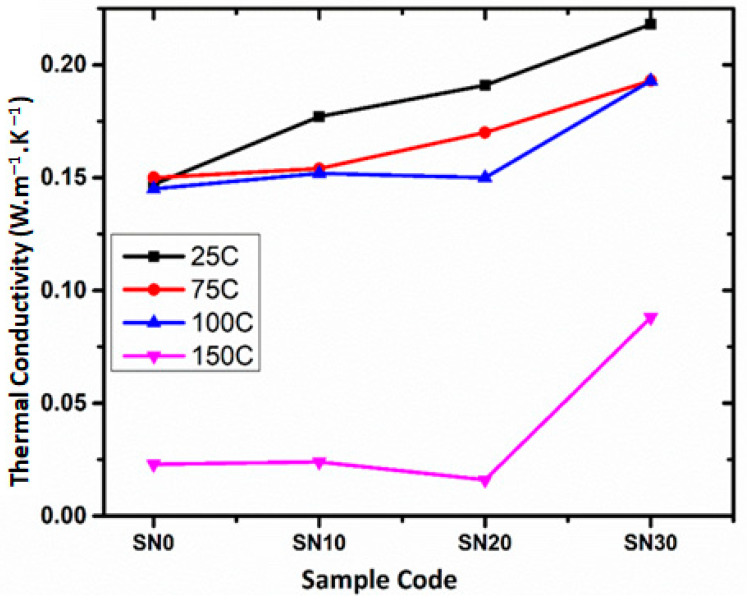
Thermal conductivity (TC) of polystyrene/m-BN composites.

**Table 1 polymers-15-00235-t001:** Sample code details.

Sample Code	Description
SN 0	Neat polystyrene
SN 10	10 wt.% loading of modified-BN in PS
SN 20	20 wt.% loading of modified-BN in PS
SN 30	30 wt.% loading of modified-BN in PS

**Table 2 polymers-15-00235-t002:** Sample codes and details of formulations.

Sample Code	(GPPS-550P)	(m-BN)	DMC	Irganox 1010	Stearic Acid
SN 0	10 g	0.0 g	0.5 g	0.1 g	0.1g
SN 10	09 g	01 g	0.5 g	0.1 g	0.1g
SN 20	08 g	02 g	0.5 g	0.1 g	0.1g
SN 30	07 g	03 g	0.5 g	0.1 g	0.1g

**Table 3 polymers-15-00235-t003:** Data of tensile strength, Young’s modulus and elongation at break for pristine PS and PS/m-BN composites.

Sample Formulation	Tensile Strength (TS) MPa	Young’s Modulus (E) GPa	Elongation at Break (E_b_) %
SN0	46.74± 0.05	2.01±0.04	11.21±0.02
SN10	47.31±0.04	2.16±0.03	10.56±0.04
SN20	54.46±0.03	3.223±0.05	4.44±0.05
SN30	57.28±0.05	3.65±0.04	2.88±0.06

**Table 4 polymers-15-00235-t004:** Thermal conductivity (TC) values of polystyrene/m-BN composites at 25, 75, 100 and 150 °C.

Sample	TC at 25 °C (W·m^−1^·K^−1^)	TC at 75 °C (W·m^−1^·K^−1^)	TC at 100 °C (W·m^−1^·K^−1^)	TC at 150 °C (W·m^−1^·K^−1^)
SN1	0.147	0.150	0.245	0.023
SN2	0.177	0.154	0.152	0.024
SN3	0.191	0.170	0.150	0.016
SN4	0.218	0.193	0.193	0.088

## Data Availability

All the data will be available to readers.

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
