# Peer review of "The Synergistic Effect of Polystyrene/Modified Boron Nitride Composites for Enhanced Mechanical, Thermal and Conductive Properties"

_polymers, 2023, doi:10.3390/polym15010235_

Round 1

Reviewer 1 Report

Dear,

The authors developed polystyrene/boron nitride nanocomposites. In my opinion, there was no formation of nanocomposites, which necessitates a review of the entire manuscript. First, the authors used a high concentration of boron nitrate (10-30% by weight), generating a traditional composite. Generally, nanocomposites are formulated with 1-5% by weight of the nanofiller. Second, the manuscript lacks traditional characterizations of nanocomposites, such as: XRD and MET. Third, from the presented SEM, boron nitride particles are large and therefore not nanoscale. In addition, the comments below should be reflected:

> At the end of the abstract present the potential application of the developed compounds;

> Add topic 1 in the manuscript - introduction. Please make clear the novelty of the manuscript in the introduction. In addition, authors must add a specific review on the topic based on the polystyrene matrix;

> Materials. For polystyrene, inform the density and melt flow index (MFI); inform the concentration of ammonium hydroxide; inform the granulometric distribution of boron nitride;

> Page 4. “Di-methacrylate (DMC) was used as a crosslinker....”. The authors tried to reticulate who? The polystyrene? If so, there is practically no possibility, polystyrene hardly cross-links, due to the aromatic ring;

> Please detail the molding process and the samples for mechanical testing in composite fabrication. Inform the molding parameters;

> FTIR (4 – a). Practically no formation of new chemical groups is observed on the surface of boron nitride. The bands are similar between pure nitride and modified nitride. The curves are the same, only a slight change in intensity;

> The authors comment on nanocomposites, but they did not perform XRD and TEM to prove the formation of nanocomposites;

> Nanocomposites are manufactured using a low concentration of nanofiller (1-5%). The authors' manuscript used boron nitride in the order of 10-30% by weight. Why nanocomposites? The manuscript developed a traditional composite;

> At the Mev did not form a nanocomposite, the authors developed composites. Furthermore, the particles are large, have no nanoscale size for nanocomposite formation;

> Page 7. “Thus, the fabricated polymeric nanocomposites presented here exhibited good mechanical.......”. The manuscript did not form a nanocomposite;

> TG. Please add a table containing: temperature for 10% mass loss; temperature for 50% mass loss; residue; add the first derivative curve;

> Figure 9. Please add the glass transition value for each composite. “It is observed from the graph that Tg of SN10, SN20 and SN30 nanocomposites shows a significant increase as compared to pristine SN0 due to the loading of m-BN”. What is the significant increase in Tg? Practically similar values around 100°C.

> “..........synergistically improves the thermal property and crystallinity of the modified nanocomposites”. Crystallinity for polystyrene? It is an amorphous polymer.

> In Table 2, the SN20 composite significantly increased the elastic modulus, in relation to SN0. Why didn't this happen in the storage module?

> Please improve the resolution of Figure 9 (a-c);

> Does thermal conductivity have no experimental error? Was the analysis performed with only one sample?

Author Response

Dear Reviewer, thank you for the comments and suggestions. Your guidance improved our knowledge regarding this research

Reviewer 2 Report

The paper presents an interesting approach based on the Synergistic effect of Polystyrene/Modified Boron Nitride Nano-composites for Enhanced Mechanical, Thermal and Conductive properties. However, the innovation of the current research work should be further highlighted and emphasized. At the same time, the authors should consider the following comments to greatly improve the quality of the paper.

1. In the abstract, add a final statement that highlights the importance of this research and its possible potentials. Also, introduce the problem in the initial lines of the abstract.

2. The introduction needs to be improved by relating to the mechanics of the studied materials and their mechanical characteristics. The references to be included are: 10.1016/j.polymertesting.2017.09.009, 10.3390/polym14132662, 10.1016/j.compstruct.2021.114698, 10.1177/0731684417727143, 10.1002/app.46770, 10.1016/j.porgcoat.2022.107015.

3. Kindly add a table that describes the main physical and chemical properties of the raw materials used in this study.

4. Were the preparation methods described by the authors come in accordance with a certain standard or do they follow previous procedures?

5. When the authors mention "After initial digestion of the ingredients at low spin", what was the spinning speed? Kindly provide full information about the twin-screw extruding process, the heating profile zones, screw geometry and final sample extraction.

7. For the tensile tests, what was the reason for the specified test conditions in this research? Do the speed of test value relate to a specific application? 

8. How many samples were used per configuration for the tensile test?

9. Table 3 represents absolute values only without any statistical insights. Kindly modify it by adding the standard deviation and variance for each sample type. Also, add a caption text to the table.

10. The conclusion needs to be modified to summarize the research outcomes in short statements with clear observations.

Author Response

Dear Reviewer, thank you for the comments and suggestions. We have incorporated the suggested recommendations and modified the manuscript.your guidance improved our knowledge greatly.

Round 2

Reviewer 1 Report

Dear, 

The authors improved the quality of the manuscript, generating greater clarity. Therefore, the manuscript has merit for publication. Before publication of the manuscript, I suggest a review in English.

Yours sincerely,

Reviewer 2 Report

The article can be accepted.